# Mn(II) Complexes of Enlarged Scorpiand-Type Azamacrocycles as Mimetics of MnSOD Enzyme

**Mario Inclán** [1,*] , **María Teresa Albelda** [2] , **Salvador Blasco** [1] , **Carolina Serena** [3] , **Javier Ugarte Chicote** [4] , **Antonio García-España** [5] **and Enrique García-España** [1]

1   Institute of Molecular Science, Universitat de València, C/Catedrático José Beltrán No. 2, 46980 Paterna, Spain; salvador.blasco@uv.es (S.B.); enrique.garcia-es@uv.es (E.G.-E.)
2   Inorganic Chemistry Department, Universitat de València, Dr. Moliner, 50, 46100 Burjassot, Spain; teresa.albelda@uv.es
3   Institut d'Investigació Sanitària Pere Virgili, Hospital Universitari Joan XXIII, 43005 Tarragona, Spain; carolserena@gmail.com
4   Pathology Unit, Joan XXIII University Hospìtal, C/Dr. Mallafrè Guasch 4, 43005 Tarragona, Spain; jugarte@piushospital.cat
5   Bionos Biotech, S.L., Biopolo Hospital La Fe, Av. Fernando Abril Martorell 106, 46026 Valencia, Spain; agarciaespana@bionos.es
*   Correspondence: mario.inclan@uv.es; Tel.: +34-963-544-377

**Abstract:** Living organisms depend on superoxide dismutase (SOD) enzymes to shield themselves from the deleterious effects of superoxide radical. In humans, alterations of these protective mechanisms have been linked to the pathogenesis of many diseases. However, the therapeutic use of the native enzyme is hindered by, among other things, its high molecular size, low stability, and immunogenicity. For this reason, synthetic SOD mimetic compounds of low molecular weight may have therapeutic potential. We present here three low-molecular-weight compounds, whose $Mn^{2+}$ complexes can mimic, at least partially, the protective activity of SOD-enzymes. These compounds were characterized by NMR, potentiometry, and, to test whether they have protective activity in vitro, by their capacity to restore the growth of SOD-deficient strains of *E. coli*. In this report, we provide evidence that these compounds form stable complexes with $Mn^{2+}$ and have an in vitro protective effect, restoring up to 75% the growth of the SOD-deficient *E. coli*.

**Keywords:** enzyme mimic; superoxide dismutase; polyamine; manganese complex; oxidative stress; antioxidant; scorpiand; scorpionate

## 1. Introduction

Enzymes play a central role in life processes as biological catalysts. Most of the essential chemical processes in living organisms, from metabolism to replication, would not be possible without them speeding up and controlling the chemical reactions that support life. Among the thousands of roles they play, there is a group of enzymes that have evolved to protect organisms from the deleterious effect of certain reactive oxygen species (ROS) produced during the metabolism of molecular oxygen. Among these ROS, superoxide radical ($O_2^-$), a metabolic by-product generated after the one-electron reduction of molecular oxygen, is recognized as one of the most toxic ones [1].

To maintain the endogenous $O_2^-$ concentration in a low, beneficial range in the nanomolar level, living organisms are equipped with a family of enzymes known as superoxide dismutase (SOD) [2]. These oxidoreductases contain either Cu, Fe, Mn or Ni in their active site, and efficiently transform $O_2^-$ into $H_2O_2$ and $O_2$. Nevertheless, a malfunctioning of these protective mechanisms in living systems results in situations of redox imbalance, known in general as oxidative stress, that have been related to a plethora

of health issues, such as cardiovascular diseases, neurodegenerative disorders, inflammation, cancer, cerebral ischemia-reperfusion injury, and even transmissible spongiform encephalopathies [3–6].

The direct treatment of this conditions with the native enzymes is limited by a series of factors, among them: the lack of oral activity, immunogenicity, high costs of production, fast degradation, and low membrane permeability. For this reason, in recent decades there have been tremendous efforts to design and prepare low-molecular-weight molecules to mimic, at least in part, the activity of SOD enzymes to be tested as drugs for the different diseases related to oxidative stress. Enzyme mimics are cheaper to produce and might have a better pharmacokinetic profile. Different ligands and metals have been tested for this purpose, and there have been several reviews published covering the topic [7–10].

Among the ligands used to prepare SOD mimetics, comprising a Mn atom as the redox centre, it is worth mentioning the success of porphyrins [11], Salen derivatives [12] and azamacrocycles. A relevant example of the later is the compound M40403, prepared by D. Salvemini et al. [13]. In the past decade, our research group has contributed to these efforts by developing several $Mn^{2+}$ complexes with polyazamacrocycles that showed high SOD activity, both in vitro and in vivo, and good pharmacokinetic characteristics [14–16]. Some of the ligands prepared belong to a particular family of polyamine macrocycles known as scorpiands, characterized by the presence of a pendant arm, attached to the macrocycle, which contains additional donor atoms [17,18]. Interestingly, $Cu^{2+}$ complexes of these ligands show negligible SOD activity. This can be explained by the large thermodynamic stability of the pentacoordinated complex they form, which prevents the metal centre from interacting with the superoxide anion and cycle between $Cu^{2+}/Cu^+$.

To increase the structural library of these scorpiand-type molecules, we prepared the compounds L1–L3 shown in Figure 1, enlarged with a propylamine chain, and therefore possessing one additional nitrogen atom in the pendant arm. Furthermore, the compounds were functionalized with different aromatic units: 2-pyridine, 2-quinoline and 1-pyrene. These substituents, of decreasing polarity, should allow us to evaluate how the overall hydrophobicity of the manganese complex affects its SOD-mimetic activity.

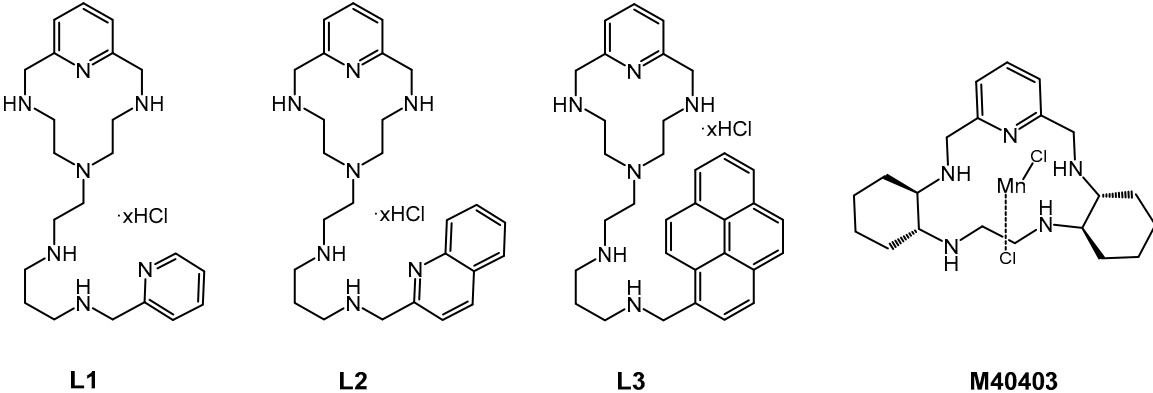

**Figure 1.** Drawing of the enlarged scorpiand-like receptors studied in this work (L1–L3) and the reference compound M40403 (see [13]).

## 2. Materials and Methods

### 2.1. Synthesis

All reagents and chemicals were obtained from commercial sources and used as received. Solvents used for the chemical synthesis were of analytical grade and were used without further purification. The ligands were prepared following the general procedures described in references [19] (tosylated precursor), [20] (L3), [21] (L1), and [22] (L2), and summarized in Figure 2. First, the pendant arm of the tosylated macrocyclic precursor was elongated using *N*-(3-bromopropyl)-phthalimide. The deprotection was performed in two steps: first the phthalimide group was removed by the Ing–Manske procedure,

reacting the compound with hydrazine and, later, the tosyl groups were removed by refluxing the compound with HBr/AcOH and phenol. Finally, the ligand, in its free amine form, was functionalized by reacting the terminal primary amine with the corresponding aldehyde. Precipitation of the product with HCl yielded the final hydrochloride salts. Characterization data ($^1$H, $^{13}$C NMR and elemental analysis) of the compounds can be found in the Supplementary Materials (Table S1). The synthesis of L3 has already been published by the authors elsewhere (see [20]).

**Figure 2.** Reaction pathway for the synthesis of L1–L3.

### 2.2. Electromotive Force Measurements (emf)

The potentiometric titrations were carried out at 298.1 K using NaCl 0.15 M as supporting electrolyte. The experimental procedure and equipment (burette, potentiometer, cell, stirrer, microcomputer, etc.) has been fully described elsewhere [23]. The acquisition of the *emf* data was performed with the computer programme PASAT [24,25]. The reference electrode was an Ag/AgCl electrode in saturated KCl solution. The glass electrode was calibrated as a hydrogen ion concentration probe by titration of previously standardised amounts of HCl, with $CO_2$-free NaOH standardized solutions, and determining the equivalent point by Gran's method [26,27], which gives the standard potential, E°′, and the ionic product of water ($pK_w = 13.73$ at 298.1 K). The computer programme HYPERQUAD [28] was used to calculate the protonation and stability constants. The pH range investigated was 2.5–11.0, and the concentration of $MnSO_4$ and of the ligands ranged from 1 to 5 mM, with a 1:1 $Mn^{2+}$:L molar ratio. The different titration curves for each system (at least two, with over 50 experimental data points each one) were treated first as separate curves and then merged and treated as a single set without significant variations in the values of the stability constants.

### 2.3. Electrochemistry

Electrochemical measurements were carried out in a device consisting of a CV-50W potentiostat and a measuring cell equipped with an argon purge and magnetic stirrer. To prepare the sample, a typical volume of 5 mL and an analyte concentration of 1 mM were used. The aqueous solutions contained 0.15 M NaCl as auxiliary electrolyte, and the pH of the solution was adjusted with HCl and NaOH. The electrodes used were Ag/AgCl as reference, platinum wire as auxiliary electrode, and a graphite working electrode.

### 2.4. NMR Measurements

The NMR spectra were recorded at 298 K using a spectrometer NMR apparatus operating at 300 MHz for $^1$H NMR spectra and at 75.43 MHz for $^{13}$C NMR spectra. Spectra

were obtained using a 5 mm inverse broadband probe head incorporating a shielded Z-gradient coil. The samples were prepared in $D_2O$ and the chemical shifts are given in ppm using 3-(trimethylsilyl)propionic-2,2,3,3-d4 acid (TSP) as a reference.

### 2.5. In Vitro Evaluation of the SOD Activity

Two strains of *E. coli* were used for these studies: a wild strain (AB1157) and an SOD-deficient strain (PN134). For growth tests, the bacteria were seeded in 96-well plates (200 μL/well), and their growth was followed turbidimetrically at 620 nm using a spectrophotometer for microtiter plates, thermostated at 37 °C. Stock solutions of all compounds in water (with $Mn^{2+}$ in a 1:1 ratio) were diluted in the culture medium (casamino acid medium, M9CA) to the desired concentration for each assay. Three independent experiments were performed and, in each of them, the compounds were tested in duplicate.

## 3. Results

### 3.1. Basicity of the Ligands and Mn(II) Coordination

The compounds prepared are polyprotic systems and it is therefore necessary to determine their protonation constants in water before attempting to calculate the stability constants with metal cations, because the proton transfer reaction will be in competition with the metal coordination. For this reason, acid–base potentiometric titrations of the ligands were performed, and the data were adjusted to a given model by a non-linear fitting procedure to obtain the global basicity constants, from which the stepwise protonation constants of L1–L2 were calculated (Table 1). The protonation constants of L3 had already been published by the authors elsewhere (see [20]).

**Table 1.** Stepwise and cumulative protonation constants of L1–L3.

| Reaction [a] | | L1 | L2 | L3 [c] |
|---|---|---|---|---|
| $L + H \rightleftarrows HL$ [b] | $K_{H1L}$ | 9.49 (1) | 9.71 (2) | 9.43 (2) |
| $LH + H \rightleftarrows H_2L$ | $K_{H2L}$ | 8.91 (1) | 9.03 (1) | 9.18 (2) |
| $H_2L + H \rightleftarrows H_3L$ | $K_{H3L}$ | 7.85 (1) | 8.00 (2) | 7.88 (3) |
| $H_3L + H \rightleftarrows H_4L$ | $K_{H4L}$ | 6.66 (2) | 6.70 (2) | 6.92 (3) |
| $H_4L + H \rightleftarrows H_5L$ | $K_{H5L}$ | | 2.25 (8) | |
| **log β** | | **32.92 (2)** | **35.69 (2)** | **33.41 (3)** |

[a] Logarithms of the stepwise ($K_{HjL}$) and cumulative (β) protonation constants, calculated as log β = $\sum_j$ log $K_{HjL}$. Values were determined at 298 K in 0.15 M NaCl. Numbers in parentheses are standard deviations in the last significant figure. [b] Charges have been omitted. [c] Values taken from [20].

As can be observed, for L1 and L3, the overall basicity, as well as the values for the stepwise protonation constants, are very close, as one would expect for structurally similar compounds. For these two ligands, four protonation constants were determined, corresponding to the protonation of the four secondary amines. In the case of the 2-quinoline substituted ligand L2, a fifth protonation step was detected, with a value of 2.25 log units, which can be ascribed to the protonation of the quinoline nitrogen atom. The overall basicity of this ligand is higher than for the other two and, as we will demonstrate later, this influences the $Mn^{2+}$ coordination properties. It was not possible to measure a fifth protonation constant for L1 or L3. This last protonation step should involve either a pyridine nitrogen atom or the central tertiary amino nitrogen of the macrocyclic ring. It is expected that the value of these protonation constants falls outside the pH interval of the technique, on account of the known lower basicity of tertiary amines in water and the electrostatic repulsion of the surrounding ammonium groups [29]. The molar distribution diagrams derived from these constants are available in the Supplementary Material (Figure S1). With this information, we can conclude that, for the free ligands and under the conditions at which the biological experiments are going to be performed (pH 7.4), the main species present in the solution, for the three ligands, is $[H_3\mathbf{L}]^{3+}$, with a prevalence of 65% for L1, 70% for L2, and 60% for L3.

The interaction of these macrocyclic ligands with $Mn^{2+}$ was then studied. The potentiometric data could only be obtained for L1 and L2, since the low solubility of $[MnL3]^{2+}$ at the concentrations required for the potentiometric technique ($\approx$1 mM) hindered the determination of the stability constant of this complex. Table 2 gathers the stability constants calculated by the best fitting of the data to a model taking into account the protonation of the ligands, as well as the hydrolysis equilibria of the metal cation. With these values, the corresponding species distribution diagrams were built (Figure 3).

**Table 2.** Stepwise stability and protonation constants for $Mn^{2+}$ complexes of L1–L3 at 298.1 $\pm$ 0.1 K and NaCl 0.15 M.

| Reaction [a] | L1 | L2 | L3 |
|:---:|:---:|:---:|:---:|
| L + Mn $\rightleftarrows$ MnL [b] | 9.67 (3) | 8.42 (3) | N.D. |
| MnL + H $\rightleftarrows$ MnHL | 6.80 (4) | 8.97 (3) | N.D. |
| MnL + $H_2O$ $\rightleftarrows$ MnL(OH) + H | - | −11.1 (1) | N.D. |

[a] Logarithms of the stability constants. Values were determined at 298 K in 0.15 M NaCl. Numbers in parentheses are standard deviations in the last significant figure. [b] Charges have been omitted.

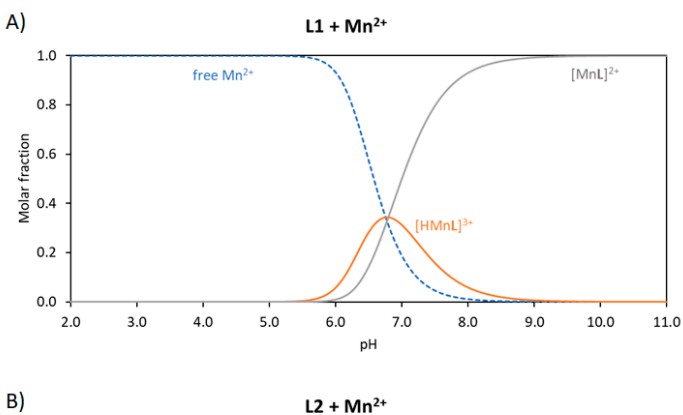

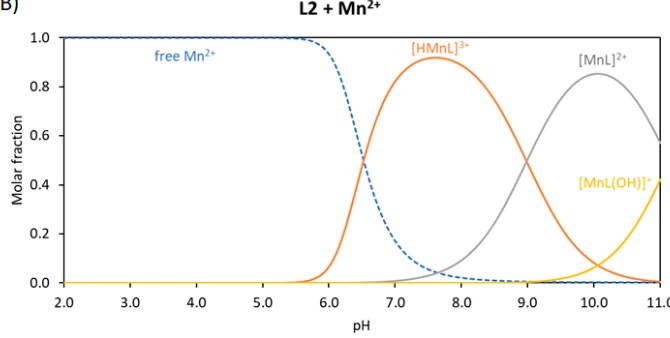

**Figure 3.** Distribution diagrams for the systems L1/$Mn^{2+}$ (**A**) and L2/$Mn^{2+}$ (**B**) in a 1:1 ratio ([L] = [$Mn^{2+}$] = 1.0 mM) as a function of pH. The dashed line represents the molar fraction of free $Mn^{2+}$.

We can estimate the overall stability of the complexes at pH 7.4 by calculating the percentage of free $Mn^{2+}$, in relation to the total amount of metal (1.0 mM), at this pH value. The results obtained are 6.1% and 6.8% for L1 and L2, respectively. This should be taken into consideration, since manganese salts might show, on their own, a certain SOD activity [30].

By looking at the distribution diagrams, we can observe a striking difference between the behaviour of these two molecules when interacting with $Mn^{2+}$. In the case of L1, the prevalent species in solution at pH 7.4 is $[Mn\mathbf{L}]^{2+}$, accounting for 75%, while for L2, under the same conditions, the main species is $[HMn\mathbf{L}]^{3+}$, with a prevalence of more than 90% over the other species. This is related to the differences in basicity described before, and will impact the SOD mimetic activity of each system.

### 3.2. Electrochemical Behaviour

For a manganese complex to act as an SOD mimetic, it is necessary that the metal be able to cycle between the $Mn^{2+}$ and $Mn^{3+}$ oxidation states. Moreover, the redox potential should be between the one-electron reduction potential for oxygen ($-160$ mV vs. NHE) and the one-electron reduction potential of superoxide radical (890 mV vs. NHE) [8]. To study and quantify this behaviour in our systems, cyclic voltammetry studies were carried out using $L1/Mn^{2+}$ as the reference. A reversible oxidation process by one electron was observed, as can be seen by the anodic (A1) and cathodic (C1) peaks that appear around +0.5 V vs. Ag/AgCl in Figure 4, in which the cyclic voltamperograms of $[Mn(L1)]^{2+}$ is represented. In this case, the relationship between the current of the cathodic peak and the current of the anodic peak is close to unity; however, it decreases with increasing scanning speed, which indicates that the initial oxidation process A1 is followed by a partial dissociation of the electrochemically generated $Mn^{3+}$. This hypothesis would explain the appearance of an additional cathodic signal near $-0.5$ V (C2), which could be attributed to the reduction of the $Mn^{3+}$ aqueous complex from the proposed dissociation process.

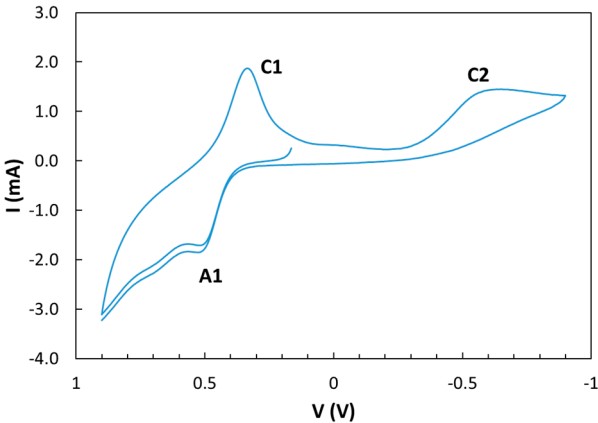

**Figure 4.** Cyclic voltamperogram of the $L1/Mn^{2+}$ system. Sweep speed = 10 mV/s.

In the complex studied, the formal potential for the $Mn^{3+}/Mn^{2+}$ pair was +736 mV (vs. SHE), which is within the required range for SOD activity. It is, however, greater than the ideal potential for oxidation and reduction of superoxide (+360 mV), being rather close to the upper limit of +890 mV. Nevertheless, this could be an advantage for enzyme mimetics since, as it has been described in the literature, higher potentials may be necessary to achieve greater selectivity for superoxide, avoiding the oxidation of $Mn^{2+}$ by other oxidizing species present in the biological medium, such as nitric oxide, peroxynitrite, hydrogen peroxide, hypochlorite, or even oxygen [31]. Moreover, the value of the formal potential of L1 is within the same range as other similar compounds built with a scorpiand-like polyaminic scaffold. For example, the six compounds reported in reference [15] had potentials in the 550–850 mV range.

### 3.3. SOD Mimetic Activity

The SOD mimetic activities of the $Mn^{2+}$ complexes of L1–L3 were tested in vitro in a bacterial model consisting of two strains of *E. coli*: a wild-type strain and an SOD-deficient mutant. The SOD-deficient strain can only grow at about 20% of the rate of the wild-type strain when cultured in a restricted media in the presence of oxygen at pH 7.4. However, an SOD-mimetic compound able to enter the bacterial cytoplasm could compensate for the SOD deficiency and increase the growth rate [32]. All three compounds showed a protective effect in the SOD-deficient strain, increasing its growth with a maximal recovery rate of about 75% for $L1/Mn^{2+}$, 65% for $L2/Mn^{2+}$ and 45% for $L3/Mn^{2+}$ at 20 µM (Figure 5). Out of the three compounds, only $L3/Mn^{2+}$ slightly reduced, by about 20%, the wild-type growth rate. In this sense, also $L3/Mn^{2+}$ in the SOD-deficient strain showed less growth at

the highest concentrations (10 and 20 µM) in comparison with lower concentrations (1 to 5 µM). Based on these results, L1 and L2 complexes seem the best candidates for future studies regarding SOD-mimetic activity.

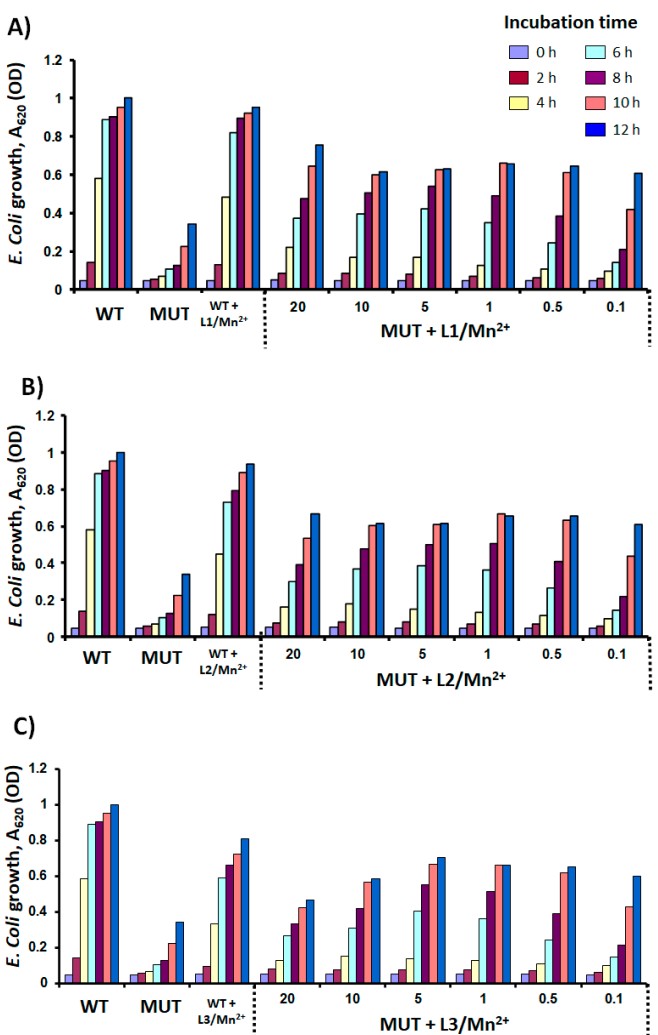

**Figure 5.** Protective effects of $L1/Mn^{2+}$, $L2/Mn^{2+}$ and $L3/Mn^{2+}$ complexes in SOD-deficient *E. coli* strain. Aerobic growth of wild-type (WT) and the sodAsodB-deficient *E. coli* (MUT) in M9CA restrictive medium incubated with 0, and 20 µM (WT), or with 0, 0.1, 0.5, 1, 5, 10 and 20 µM (MUT) concentrations of $L1/Mn^{2+}$ (**A**), $L2/Mn^{2+}$ (**B**), or $L3/Mn^{2+}$ (**C**) complexes. Bacterial growth was monitored at 620 nm and plotted in the graph at 0, 2, 4, 6, 8, 10 and 12 h of incubation (see figure inset). The graph is a representative determination of three independent experiments done in duplicate.

Control experiments with the complexes were performed against the wild-type strain to test the toxicity at the highest concentration. As can be seen in Figure 5, only $L3/Mn^{2+}$ lowers the rate of growth to a significant extent (about −20%). Control experiments with the free ligand and the mutant strain showed no significant increase of the growth rate at the highest concentrations. The possible effect of the free $Mn^{2+}$ was considered negligible at those concentrations based on previous experiments.

## 4. Conclusions

Ligands L1 and L2 form stable complexes with $Mn^{2+}$ in aqueous media at pH values higher than 7.0. The complexes formed by L3 were too insoluble to be characterized by the potentiometric technique. In the case of $L1/Mn^{2+}$, the formal potential for the $Mn^{3+}/Mn^{2+}$ pair falls well within the optimal interval for the dismutation of superoxide. When tested

against an SOD-deficient mutant strain of *E. coli* the three $Mn^{2+}$ complexes essayed showed a relevant increase in the growth rate, with the complex of the 2-pyridine substituted ligand L1 being the most efficient. L3 will not be considered for future studies, since it showed some toxicity at higher doses.

**Supplementary Materials:** The following supporting information can be downloaded at: https://www.mdpi.com/article/10.3390/app12052447/s1, Table S1. Characterization data ($^1$H and $^{13}$C NMR and Elemental Analysis) of L1–L3, Figure S1. Molar distribution diagrams of the different protonated species.

**Author Contributions:** Writing—original draft preparation, M.I.; methodology, validation, investigation, writing—review and editing, M.I., M.T.A., S.B., C.S., J.U.C., A.G.-E., E.G.-E. All authors have read and agreed to the published version of the manuscript.

**Funding:** We acknowledge the financial support by the Spanish MINECO and FEDER funds from the European Union (Projects PID2019-110751RB-I00, CTQ2017-90852-REDC and Unidad de Excelencia María de Maeztu CEX2019-000919-M). Mario Inclán thanks Generalitat Valenciana and the European Social Fund for the financial support (APOSTD2019/159).

**Institutional Review Board Statement:** Not applicable.

**Informed Consent Statement:** Not applicable.

**Conflicts of Interest:** The authors declare no conflict of interest.

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
