# Peer review of "Mn(II) Complexes of Enlarged Scorpiand-Type Azamacrocycles as Mimetics of MnSOD Enzyme"

_applsci, doi:10.3390/app12052447_

Round 1
Reviewer 1 Report
The research describes plausible supramolecular systems, synthetically affordable and already proved to have an overall good safety profile, to be used as Superoxide Dismutase (SOD) mimetics in an interesting, novel approach. The ability of these systems to form stable complexes with metal cations i.e. Mn2+ has been proved potentiometrically and their ability to restore the SOD-activity in SOD-lacking mutant E. Coli strains has been successfully tested in vitro.
The focus of the article is well-described in the introduction thanks to a thoroughly compiled piece of relevant up-to-date literature, from which it is highlighted the rooted expertise of the group aiming to publish the work in the field of interest. The research approach is scientifically sound and they used well-established methodologies, correctly implemented, and carefully presented and interpreted throughout the manuscript.
The research article is robust and complete, perfectly targeting the special issue of the journal in which it aims to be published. A more complete and interesting scenario could have been described if more metal centres would have been tested, since different types of SOD enzymes interact with different cations. Moreover, also the ligands have been shown to interact with other suitable metals i.e. Cu, Fe, Ni.
The formation of supramolecular complexes might have been tested and deepend with other techniques, i.e. HPLC, ITC, MS spectrometry, in silico. For the in vitro experiments, a blank test with non-complexed ligands could have been inserted/cited in order to discrimate between eventual toxicity of the compounds themselves preventing the bacterial (re-)growth. These suggestions does not affect the relevance of the presented work which can be accepted and published after minor revision. The following points should be adressed:
MAIN BODY
- Line 22: “they have protective activity in vitro”.
- Line 24: “and have an in vitro protective”.
- Line 27: “scorpionate”.
- Line 55: Why Mn has been chosen and not other metal cations among the ones cited above in the introduction (line 40: Cu, Fe, Ni)? Is it a matter of previously tested affinity with the ligands or it is driven by the characteristics of the SOD? Please, specifiy better on this point.
- Line 81: Reference 19 does not describe the synthesis of the molecules of interest, it can be removed. Reference 20 and 21 describe the synthesis of L3 and L1, respectively. Reference 16, instead, describes the synthesis of L2.
- Line 82: Is the tosylated macrocyclic precursor available commercially or synthesized?
- Line 88: In the text, compounds are described as “hydrochloride salts”. This is not mirrored in Figure 2, where the structures are depicted with non-protonated, free amine groups. Please, update Figure 2.
- Line 95: “2. Electromotive force (emf) measurements” Abbreviation “emf” appears at line 98.
- Line 97: “The experimental procedure and equipment”.
- Line 122: “The chemical shifts are given in ppm relative to the solvent signal.” The NMR signals are given using D2O peak as reference, but relative to standard TMS? Please, explain this better.
- Table 1: Labels for the stepwise constants (KHjL) can be inserted in each line of Table 1 for better clarity as the cumulative constant label (β) appears in the last line.
- Line 149: “with a value of 2.23 log units”. Value 2.23 does not match with the value 2.25 in Table 1.
- Line 161: “[H3L]3+.
- Line 162: Which salt of Mn2+ has been used? It is worth to mention, at least once in the materials and methods, since the counterions can affect the water solubility.
- Table 2: Again, labels for the stepwise constants (KHjL) can be inserted in Table 2, as for Table 1. In this case, also the cumulative constants (β) are missing.
- Figure 3: In the caption, the labels A and B are missing, while upper and lower panels are used.
- Line 187: In section “3.2. Electrochemical behaviour”, L2 is not mentioned at all. Is there a particular reason not to study the elecrochemical response of L2-Mn complex? L1 was considered as reference?
- Line 191: Reference 8 report the values of -0.16 V and 0.89 V vs NHE, not vs SHE as written.
- Line 206: The values +736 mV, +360 mV (line 208) and +890 mV (line 209) are expressend in mV while the reference values (-0.16 V and 0.89 V) are expressed in V. Please, select the unit that best fits the case.
- Line 216: In section “3.3. SOD mimetic acitivity”, blank experiments could have been performed by administering L1/L2/L3 alone to the bacterial strains as well as Mn2+ alone, in the concentration expected from data riported at line 178 (6.1% and 6.8%). Is there any data that can be added or cited? However, as riported in reference 28, the amount of free cation and thus its SOD-activity is probably negligeable.
- Line 217: “were tested in vitro”.
- Line 220: “when cultured in restricted media in the presence of oxygen at pH = 7.4”. Info about pH value was mentioned at line 160.
- Figure 5: i) Also in this caption, the labels A, B and C are missing, only upper, middle and lower panels are used.
ii) In the three graphs, the x-axis label WT+L might be corrected into WT+L/Mn2+ as for MUT+L/Mn2+.
REFERENCES
- Line 285: Reference 12 is given as part of a book. If the authors agree, the reference can be adjusted to the corresponding article, easier to access (https://doi.org/10.1016/S1054-3589(08)60987-4): R. Doctrow, K. Huffman, C.B. Marcus, W. Musleh, A. Bruce, M. Baudry, B. Malfroy, Salen-Manganese Complexes: Combined Superoxide Dismutase / Catalase Mimics with Broad Pharmacological Efficacy. Adv. Pharmacol 1996, 247–269.
- Line 312: Reference 23 could not be found online. Please, if available insert a link.
- Line 315: “Gans, P.; Sabatini, A.; Vacca, A.”. Order of authors is inverted.
SUPPLEMENTARY MATERIALS
- Table S1: i) The word NMR is misspelled in the 1H and 13C characterizations.
ii) The integration of 1H signals for L3 should be checked (31 Hs instead of expected 36 Hs) .
Author Response
We would like to thank the Reviewers for their thorough revision and useful suggestions, which have been very helpful in improving the manuscript.
Here are the answers, point by point, to the suggestions that the reviewers have made. All changes in the manuscript have been marked up using the “Track Changes” function.
Best regards.

Reviewer 2 Report
Dear authors manuscript entitled Mn(II) complexes of enlarged scorpiand-type azamacrocycles as mimetics of MnSOD enzyme is interesting with moderate novelty impact. The manuscript is well written with attention to detail. The suggestion that could help with improving the manuscript are listed below.
Comments;
- Line 55-71. Figure 1 should be rather moved to materials and methods sections or thoroughly removed. In the marked part of the manuscript, the authors should present a clear working hypothesis.
- Footnotes of table 1; Values taken from reference Error! Bookmark not defined. Please correct.
- Line 214-215; Authors could report an exact range from mentioned reference and cite it correctly without “see, for example, ref.15”
- Authors perform an assay of SOD mimetic activity with the use of bacterial cells as models. Why the simple comparison with an easily available commercial preparation of microbial origin MnSOD enzyme was not performed?
Author Response

(The authors gave the same response as above.)
